# Self-Attention Diffusion Models for Zero-Shot Biomedical Image Segmentation: Unlocking New Frontiers in Medical Imaging

**DOI:** 10.3390/bioengineering12101036

**Published:** 2025-09-27

**Authors:** Abderrachid Hamrani, Anuradha Godavarty

**Affiliations:** Optical Imaging Laboratory, Department of Biomedical Engineering, Florida International University, 10555 West Flagler Street, EC 2675, Miami, FL 33174, USA

**Keywords:** medical image segmentation, zero-shot learning, unsupervised learning, self-attention mechanisms, diffusion models, deep learning, generative models

## Abstract

Producing high-quality segmentation masks for medical images is a fundamental challenge in biomedical image analysis. Recent research has investigated the use of supervised learning with large volumes of labeled data to improve segmentation across medical imaging modalities and unsupervised learning with unlabeled data to segment without detailed annotations. However, a significant hurdle remains in constructing a model that can segment diverse medical images in a zero-shot manner without any annotations. In this work, we introduce the attention diffusion zero-shot unsupervised system (ADZUS), a new method that uses self-attention diffusion models to segment biomedical images without needing any prior labels. This method combines self-attention mechanisms to enable context-aware and detail-sensitive segmentations, with the strengths of the pre-trained diffusion model. The experimental results show that ADZUS outperformed state-of-the-art models on various medical imaging datasets, such as skin lesions, chest X-ray infections, and white blood cell segmentations. The model demonstrated significant improvements by achieving Dice scores ranging from 88.7% to 92.9% and IoU scores from 66.3% to 93.3%. The success of the ADZUS model in zero-shot settings could lower the costs of labeling data and help it adapt to new medical imaging tasks, improving the diagnostic capabilities of AI-based medical imaging technologies.

## 1. Introduction

Biomedical image segmentation serves a crucial role in healthcare, facilitating diagnosis, treatment planning, and disease monitoring. Convolutional neural networks (CNNs), particularly the U-Net architecture, demonstrate effectiveness in this domain by utilizing a contracting path to capture context and an extensive path for precise localization [1,2]. These networks excel at biological image segmentation and cellular tracking across various microscopy modalities [3]. The success of deep convolutional networks stems from extensive labeled datasets like ImageNet [4,5]. However, biomedical image segmentation faces significant challenges due to the scarcity of annotated data [6,7]. While data augmentation techniques expand training datasets, they generate variations in existing samples without introducing true biological diversity. These synthetic modifications cannot capture the full spectrum of anatomical variations, pathological manifestations, and imaging conditions present in clinical settings [8].

Recent research explores unsupervised and zero-shot learning approaches to address these limitations [9,10,11,12]. Stable diffusion models exhibit the ability to learn inherent object concepts within their attention layers, enabling high-quality segmentation without extensive labeled datasets [13]. This capability proves particularly valuable in biomedical imaging, where manual annotation requires expert knowledge and significant time investment. The complexity of biomedical images varies significantly due to imaging modality, patient anatomy, and pathological variations, necessitating automated segmentation methods that operate with minimal supervision. Self-attention diffusion models present a promising solution [14,15]. These models excel at learning complex data distributions and capturing fine-grained details through their self-attention mechanisms. The ability to discern long-range dependencies and contextual information makes them particularly suited for biomedical image segmentation tasks [16,17].

This study introduces ADZUS (Attention Diffusion Zero-shot Unsupervised System), a deep learning model that leverages self-attention diffusion mechanisms for zero-shot biomedical image segmentation. The model generates precise segmentation masks through iterative merging of attention maps based on their Kullback–Leibler (KL) divergence [18,19], enabling coherent region identification while reducing computational redundancy. Our contributions include demonstrating the effectiveness of self-attention diffusion models for zero-shot biomedical image segmentation, achieving superior performance on benchmark datasets, and eliminating dependence on annotated data. This approach significantly reduces barriers to deploying advanced segmentation models in resource-constrained settings. Extensive experiments across various biomedical datasets validate the model’s versatility and robustness.

### Motivation

Stable Diffusion models, initially designed for image generation, have demonstrated the capability to produce highly realistic and detailed images using only text prompts. Figure 1 illustrates this potential by comparing images generated by Stable Diffusion with real clinical cases across three different imaging modalities: white blood cells, skin lesions, and chest X-rays. The top row (a, b, c) presents synthetic images generated using descriptive text prompts in [20], while the bottom row (d, e, f) displays corresponding real medical cases.

To generate these images, Stable Diffusion was prompted with medical descriptions such as


*“A highly detailed, realistic microscopic view of white blood cells, with clear distinction of nuclei and surrounding red blood cells, captured under high-resolution clinical lighting.”*

*“A close-up of a malignant skin lesion with irregular borders, dark pigmentation, and slight ulceration, presented in high-resolution medical imaging detail.”*

*“A high-resolution X-ray scan of a human chest, capturing clear details of the ribcage, lungs, and mediastinum, optimized for clinical radiographic examination.”*


The synthetic images closely resemble real medical cases taken from databases in [21,22,23,24,25,26,27,28], accurately capturing critical features such as cellular morphology in microscopic imaging, pigmentation patterns in skin lesions, and structural details in chest radiographs. This high degree of realism suggests that the self-attention mechanisms within Stable Diffusion models inherently capture relevant medical imaging concepts, making them promising candidates for zero-shot segmentation tasks.

Given this capability, an important question arises: has Stable Diffusion been trained on medical images, particularly those used in this study? To address this, we conducted a data similarity verification analysis using the publicly available LAION-5B search tool, which forms the primary training dataset of Stable Diffusion. The images used in this study, including those in Figure 1d–f and those in the results section, were tested against the LAION-5B database to determine if they were present in Stable Diffusion’s pretraining corpus. Our analysis revealed no identical or closely matching images within the LAION-5B dataset. These findings confirm that our segmentation experiments were conducted without prior model exposure to the images used in this study, ensuring a fair evaluation of ADZUS for zero-shot medical image segmentation.

## 2. Methodology

A pre-trained stable diffusion model is leveraged by ADZUS, with utilization of its self-attention layers to generate high-quality segmentation masks. In Section 2.1, a concise overview of the stable diffusion model architecture will be provided, followed by a detailed introduction to ADZUS in Section 2.2.

### 2.1. Overview of the Stable Diffusion Model

The stable diffusion model [13], a well-known variant within the diffusion model family [29,30], is a generative model that operates through both forward and reverse passes. During the forward pass, Gaussian noise is incrementally added at each time step until the image becomes completely isotropic Gaussian noise. Conversely, in the reverse pass, the model is trained to progressively eliminate this Gaussian noise, thereby reconstructing the original clean image. The stable diffusion model [13] incorporates an encoder–decoder and U-Net architecture with attention layers [29].

Initially, an image x∈ℝH×W×3  is compressed into a latent space with reduced spatial dimensions z∈ℝh×w×c  using an encoder z=Ex . This latent space can then be decompressed back into the image x˜=Dz  through a decoder. All diffusion processes occur within this latent space via the U-Net architecture, which is the primary focus of this paper’s investigation. The U-Net consists of modular blocks, including 16 specific blocks composed of ResNet layers and Transformer layers (Figure 2). The Transformer layer uses two attention mechanisms: self-attention to learn global attention across the image and cross-attention to learn attention between the image and optional text input.

The component of interest for our investigation is the self-attention layer in the Transformer layer. Specifically, there are 16 self-attention layers distributed across the 16 composite blocks, resulting in 16 self-attention tensors. Each attention tensor Ak∈ℝhk×wk×hk×wk  is 4-dimensional. Inspired by DiffuMask [31], which demonstrates object grouping in the cross-attention layer, it is hypothesized that the unconditional self-attention also contains inherent object grouping information, which can be used to produce segmentation masks without text inputs.

For each spatial location (*I, J*) in the attention tensor, the corresponding 2D attention map AkI,J,:,:∈ℝhk×wk  captures the semantic correlation between all locations and the location (*I, J*). Each location (*I, J*) corresponds to a region in the original image pixel space, the size of which depends on the receptive field of the tensor.

Two important observations motivate the method proposed in the next section:

Intra-Attention Similarity: Within a 2D attention map $A_{k}\left[I,J,:,:\right]$, locations tend to have strong responses if they correspond to the same object group as (*I, J*) in the original image space.Inter-Attention Similarity: Between two 2D attention maps, e.g., $A_{k}\left[I,J,:,:\right]$ and $A_{k}\left[I+1,J+1,:,:\right]\;$, they tend to share similar activations if (*I, J*) and (*I + 1, J + 1*) belong to the same object group in the original image space.

The resolution of the attention map dictates the size of its receptive field concerning the original image. Lower resolution maps (e.g., 8 × 8) provide better grouping of large objects, while higher resolution maps (e.g., 16 × 16) offer more fine-grained grouping of components within larger objects, potentially identifying smaller objects more effectively. The current stable diffusion model has attention maps in four resolutions: 8 × 8, 16 × 16, 32 × 32, and 64 × 64. Building on these observations, a simple heuristic is proposed to aggregate weights from different resolutions and an iterative method to merge all attention maps into valid segmentation masks. In our experiments, the stable diffusion pre-trained models from “Huggingface” are used [32]. Typically, these prompt-conditioned diffusion models run for 50 or more diffusion steps to generate new images. However, to efficiently extract attention maps for an existing clean image without conditional prompts, we use only the unconditioned latent and run the diffusion process once. The unconditional latent is calculated using an unconditioned text embedding. We set the time-step variable *t* to a large value (e.g., *t = 300*) so that real images are viewed as primarily denoised generated images from the diffusion model’s perspective.

### 2.2. ADZUS Model

Since the self-attention layers capture inherent object grouping information in spatial attention (probability) maps, we propose ADZUS, a simple post-processing method, to aggregate and merge attention tensors into a valid segmentation mask. As shown in Figure 3, the pipeline consists of three components: attention aggregation, iterative attention merging, and non-maximum suppression. ADZUS is built on pre-trained stable diffusion models. For our implementation, we use stable diffusion V1.4 [13].

#### 2.2.1. Attention Aggregation

Given an input image passing through the encoder and U-Net, the stable diffusion model generates 16 attention tensors. Specifically, there are 5 tensors for each of the dimensions: (64 × 64 × 64 × 64), (32 × 32 × 32 × 32), (16 × 16 × 16 × 16), and (8 × 8 × 8 × 8). The goal is to aggregate attention tensors of different resolutions into the highest resolution tensor. To achieve this, the last 2 dimensions of all attention maps are up-sampled (bilinear interpolation) to 64 × 64, their highest resolution. Formally, for Ak∈ℝhk×wk×hk×wk :(1)A˜k=Bilinear-upsampleAk∈ℝhk×wk×64×64 

The first 2 dimensions indicate the locations to which attention maps are referenced. Therefore, we aggregate attention maps accordingly. For example, as shown in Figure 4, the attention map in the (0, 0) location in c is first upsampled and then repeatedly aggregated pixel-wise with the 4 attention maps (0, 0), (0, 1), (1, 0), (1, 1) in Az∈ℝ16×16 . Formally, the final aggregated attention tensor Af∈ℝ64×64  is:(2)AfI,J,:,:=∑k∈1, …, 16A˜kI/δk,J/δk,:,:∗Rk
where δk=64/wk  and ∑kRk=1 . The aggregated attention map is normalized to ensure it is a valid distribution. The weights ***R*** are important hyper-parameters and are proportional to the resolution wk .

#### 2.2.2. Iterative Attention Merging

In this step, the algorithm computes an attention tensor Af∈ℝ64×64 . The goal is to merge the 64 × 64 attention maps in the tensor Af  to a stack of object proposals where each proposal likely contains the activation of a single object or category. Instead of using a K-means algorithm, which requires specifying the number of clusters, we generate a sampling grid from which the algorithm can iteratively merge attention maps.

A set of M×M  evenly spaced anchor points are generated. We then sample the corresponding attention maps from the tensor Af . This operation yields a list of M2  2D attention maps as anchors:(3)La=Afim,jm,:,: ∈ ℝ64×64im,jm∈M

To measure similarity between attention maps, we use KL divergence:(4)$$\begin{array}{l} 2\;\times \;D\left(A_{f}\left[i,j\right],\;A_{f}\left[y,z\right]\right)=\\ \left(\mathrm{KL}\left(A_{f}\left[i,j\right]\left\|A_{f}\left[y,z\right]\right.\right)+\mathrm{KL}\left(A_{f}\left[y,z\right]\left\|A_{f}\left[i,j\right]\right.\right)\right) \end{array}$$

We start with N iterations of the merging process, where we compute the pair-wise distance between each element in the anchor list and all attention maps, averaging all attention maps with a distance smaller than a threshold τ . This process is repeated in subsequent iterations, reducing the number of proposals by merging maps with distances smaller than τ . A detailed sensitivity analysis of the KL divergence threshold (τ) is provided in Appendix A.

To consolidate the multiple attention maps into coherent object-level proposals, we employ an Iterative Attention Merging algorithm. This procedure progressively merges spatially and semantically similar attention maps, guided by KL-divergence similarity, until stable segmentation proposals are formed. The detailed steps are summarized in Algorithm 1.
**Algorithm 1:** Iterative Attention Merging**Required parameters: **La, Af, N, τLa=1ν∑i, j∈νAfi, jυ=1, …, M2  **where**  v=i,jDLaυ, Afi,j<τ**for** *N-1* iterations **do**    Initialize L˜p=[]   **for** A  in Lp
**do**       V=1ν∑υ∈νLpυ # Merge attention maps          where  v=υDA,Lpυ<τ      Add V  to L˜p      Remove Lpυ ∀υ∈v from Lp    **end for**       Lp←L˜p**end for**

#### 2.2.3. Non-Maximum Suppression

The iterative attention merging step yields a list Lp ∈ ℝNp×64×64  of Np  object proposals in the form of attention maps. To convert the list into a valid segmentation mask, we use non-maximum suppression (NMS). Each element is a probability distribution map, and the final segmentation mask S ∈ ℝ512×512  is obtained by upsampling all elements in Lp  to the original resolution and taking the index of the largest probability at each spatial location across all maps. This methodology, combining attention aggregation, iterative merging, and non-maximum suppression, forms the core of the ADZUS approach for producing high-quality segmentation masks.

The iterative attention merging step yields a list of object proposals in the form of attention maps. To convert this list into a valid segmentation mask, we apply NMS, ensuring the selection of the most relevant segmented regions. Rather than directly identifying a specific anatomical structure, ADZUS generates a comprehensive segmentation mask that delineates multiple regions within the image, leveraging its self-attention mechanisms to outline the boundaries of all distinguishable structures. The model performs guided segmentation, generating a set of segmented regions without inherently classifying or labeling a specific area. Instead, it provides a structured segmentation output, allowing clinicians or users to interactively select the relevant region of interest based on the specific medical application.

## 3. Results

This section presents the evaluation of ADZUS across a range of medical image segmentation tasks to assess its zero-shot segmentation capabilities. The performance of ADZUS is benchmarked against state-of-the-art deep learning models in skin lesion segmentation, chest X-ray infection segmentation, and white blood cell segmentation. These tasks cover diverse imaging modalities, including dermoscopic images, wound photography, radiographic images, and microscopy, highlighting ADZUS’s versatility. Both quantitative metrics, such as dice similarity coefficient (DSC), intersection over union (IoU), precision, and recall, and qualitative visualizations are provided to demonstrate the model’s effectiveness in accurately delineating structures without the need for labeled training data. A detailed description of performance metrics used in this analysis is provided in Appendix A.

### 3.1. Skin Lesion Segmentation

In this section, the international skin imaging collaboration (ISIC) datasets from 2016 and 2017 have been utilized to provide a comprehensive and diverse benchmark for skin lesion segmentation. The ISIC 2016 dataset [21] included 900 dermoscopic lesion images for training and 379 images with ground truth annotations for testing, focusing on binary segmentation tasks to detect melanoma. The ISIC 2017 dataset [22] expanded the scope by providing 2000 dermoscopic lesion images for training and 600 images with ground truth masks for testing. For the evaluation of the ADZUS model, only the test data from these datasets are used, as ADZUS operates in a zero-shot manner and does not require training. The table below (Table 1) compares the performance of ADZUS with other segmentation models on the ISIC datasets 2016 and 2017.

The results in the table demonstrate that ADZUS achieved competitive performance compared to state-of-the-art segmentation models across the ISIC 2016 and 2017 datasets. In the ISIC 2016 dataset, ADZUS outperformed the top evaluation from the U-Net implementation, achieving a DSC score of 92.9% and an IoU score of 86.8%, compared to 91% and 84.3%, respectively. For the ISIC 2017 dataset, ADZUS achieved a DSC score of 88.7% and an IoU score of 80%, closely aligning with the performance of advanced models such as the EfficientNet-Based Model, which achieved a DSC score of 88% and an IoU score of 80.7%.

Figure 5 illustrates the qualitative performance of the ADZUS model on the ISIC 2016 and ISIC 2017 datasets. In both ISIC 2016 and 2017, the predicted boundaries (green) closely match the true boundaries (red), achieving high DSC scores (from 0.98 to 0.84). These results highlight ADZUS’s effectiveness as a zero-shot segmentation model for skin lesion boundaries.

### 3.2. Diabetic Foot Wound Segmentation

Building on the performance of ADZUS in skin lesion segmentation, this section transitions to evaluate its effectiveness in diabetic foot wound segmentation. To this end, ADZUS is compared against state-of-the-art segmentation models using the publicly available chronic wound dataset [37]. This dataset consists of 1010 labeled images of diabetic foot ulcers, with 810 images designated for training and 200 for inference. It provides a standardized platform for segmentation performance assessment within a supervised learning framework. The state-of-the-art models includes LinkNet-EffB1 + UNet-EffB2 [38,39], DeepLabV3Plus [40,41], Swin-Unet [39], DDRNet [39], SegFormer-b5 [42], and FUSegNet [39], which were used for the comparison.

As illustrated in Table 2, ADZUS demonstrates competitive performance (using only the test dataset of 200 images) when compared with state-of-the-art segmentation models. Notably, ADZUS outperforms the previously best-performing FUSegNet model by achieving the highest IoU of 86.68% compared to FUSegNet’s IoU of 86.40%. In terms of DSC score, ADZUS ranked slightly behind DeepLabV3Plus, LinkNet-EffB1 + UNet-EffB2 and FUSegNet (which achieved the highest DSC score of 92.70%). These findings collectively highlight the strength of ADZUS in delivering accurate and reliable segmentation results without the need for labeled training data, a significant advantage over supervised models such as FUSegNet and SegFormer-b5.

The qualitative results for the chronic wound dataset are illustrated in Figure 6. The segmentation outputs demonstrate the performance of ADZUS in accurately delineating wound regions, with original boundaries depicted in red and predicted boundaries in green. Additional qualitative results for the chronic wound dataset are presented in Figure 7, offering a comparative analysis of segmentation outputs from ADZUS and benchmark models. The figure highlights ADZUS’s ability to achieve precise segmentation, closely aligning with the original boundaries and minimizing deviations. With a DSC score of 93.56%, ADZUS demonstrates competitive performance, surpassing supervised models like FUSegNet and SegFormer, while performing slightly below DeepLabV3+.

### 3.3. Chest X-Rays Segmentation for COVID Dataset

In this section, we evaluate the ADZUS model for infection segmentation in chest X-ray images. This task focuses on identifying and segmenting infected regions in lung radiographs, which is crucial for diagnosing and monitoring respiratory diseases such as COVID-19 and pneumonia. The dataset used consists of 2913 chest X-ray images (583 images for test) [23,24,25,26].

Table 3 shows that ADZUS demonstrates superior performance in infection segmentation in chest X-ray images, outperforming state-of-the-art models across all evaluation metrics. ADZUS achieved the highest IoU (66.3%), precision (77.9%), and recall (83.1%), surpassing established deep learning architectures such as ResNet18, ResNet50, EfficientNet-b0, MobileNet_v2, and DenseNet121. Notably, ADZUS outperformed the best-performing DenseNet121 model in terms of IoU and precision scores, emphasizing its ability to accurately delineate infection regions with minimal false positives. The improved recall score suggests that ADZUS is particularly effective at capturing infected lung regions, reducing the likelihood of under-segmentation compared to traditional CNN-based approaches.

The qualitative results for infection segmentation in chest X-ray images are illustrated in Figure 8. The segmentation outputs demonstrate the performance of ADZUS in accurately delineating infection regions, with ground truth boundaries shown in red and predicted boundaries depicted in green. The figure emphasizes ADZUS’s ability to generate high-fidelity segmentation masks across different cases, achieving DSC of 90.39%, 84.94%, and 75.52%.

### 3.4. White Blood Cell Segmentation

The robustness of ADZUS is further demonstrated through its performance on the WbcMSBench dataset [26,27,28], a microscopy imaging dataset consisting of 80 test samples. Designed for multi-class segmentation across three distinct classes, WbcMSBench presents challenges due to the high variability in white blood cell morphology and overlapping structures. Table 4 shows that ADZUS achieves an IoU of 93.3%, precision of 96.5%, and recall of 96.7%, performing competitively with state-of-the-art models such as DenseNet121 (IoU: 93.8%) and EfficientNet-b0 (IoU: 93.7%). While ADZUS slightly trails these top-performing models in IoU, it matches them in precision and recall, highlighting its effectiveness in accurately delineating cellular boundaries and distinguishing between different cell classes. Figure 9 shows the qualitative results of ADZUS on the WbcMSBench dataset [26] for multi-class white blood cell segmentation. The predicted masks closely align with the ground truth, demonstrating ADZUS’s high accuracy across different cell types. All classes show strong performance, emphasizing the model’s robustness in handling complex cellular structures.

## 4. Discussion

The results presented in this study demonstrate the effectiveness of ADZUS as a zero-shot medical image segmentation model across multiple imaging modalities. ADZUS achieved competitive performance in skin lesion segmentation, chest X-ray infection segmentation, diabetic foot wound segmentation, and white blood cell segmentation, often surpassing or closely aligning with state-of-the-art supervised models.

In skin lesion segmentation, ADZUS outperformed the top U-Net implementation on the ISIC 2016 dataset, achieving a higher DSC score and IoU. On the ISIC 2017 dataset, its performance was slightly below the EfficientNet-based model in IoU, although ADZUS matched its DSC score. This outcome highlights ADZUS’s robustness in handling diverse lesion types, from melanomas to seborrheic keratosis, while also revealing its limitations when datasets exhibit high inter-class similarity. Such conditions can make fine distinctions between lesion types more challenging, which in turn impacts segmentation quality.

In the diabetic foot wound segmentation task, ADZUS exceeded the performance of the previously best-performing FUSegNet model in IoU and precision, despite operating in a completely unsupervised manner. This finding is particularly significant, as supervised models leverage extensive labeled data to optimize performance, whereas ADZUS delivers comparable accuracy without such data. However, its slightly lower DSC score compared to FUSegNet suggests that ADZUS may struggle with fine-grained boundaries in complex wound structures. We provide additional qualitative examples (Figure 6 and Figure 7) that illustrate these cases, showing both the strengths and the areas where segmentation completeness can be improved. ADZUS demonstrated superior performance in chest X-ray infection segmentation, outperforming architectures like DenseNet121 and EfficientNet-b0 across all key metrics. The model’s high recall indicates its strength in accurately capturing infected lung regions, minimizing the risk of under-segmentation, which is critical in clinical contexts such as COVID-19 diagnosis and monitoring. This result underscores ADZUS’s ability to handle macro-level infection patterns despite the variability and complexity inherent in radiographic images. In white blood cell segmentation, ADZUS performed competitively with state-of-the-art models such as DenseNet121 and EfficientNet-b0, achieving high IoU, precision, and recall. The qualitative results confirm that ADZUS effectively distinguishes between different cell types in multi-class segmentation tasks, handling challenges posed by overlapping structures and varying cell morphology. This performance further validates the model’s adaptability to microscopic imaging applications, extending its relevance beyond macroscopic medical imaging.

While ADZUS proves effective across a range of tasks, certain limitations warrant further exploration. Specifically: (1) Performance tends to be lower in datasets with high inter-class similarity, as observed in ISIC 2017. (2) ADZUS may also underperform when precise, fine-grained boundaries are required, as in some wound images. Addressing these challenges represents a promising avenue for future research. Semi-supervised fine-tuning or domain-specific adaptations could enhance performance in such scenarios. By leveraging the text-to-image generation capabilities inherent in stable diffusion models, ADZUS could also be extended to incorporate text-guided segmentation, where natural language descriptions of anatomical structures or pathological features guide the segmentation process. This enhancement would allow medical professionals to refine segmentation boundaries and identify specific regions of interest using descriptive text prompts.

In future work, ADZUS will benefit from expanding its evaluation to additional medical imaging modalities, such as retinal vessel segmentation or organ segmentation in CT scans, to further validate its generalizability. The text-guided approach could be particularly valuable in complex settings where precise anatomical descriptions can help distinguish between visually similar structures. Moreover, exploring the integration of self-supervised learning techniques, domain adaptation strategies, and runtime optimization will further improve its performance and feasibility in clinical environments. In addition, we report on computational feasibility. As noted previously, the use of a large stable diffusion backbone and iterative merging steps increases computational overhead, preventing the method from achieving strict real-time performance. Nevertheless, by requiring only a single forward pass through the diffusion model and a limited number of merging iterations, ADZUS remains practical for near real-time clinical use. Future work will focus on optimizing runtime, such as reducing anchor density, parallelizing attention merging, or leveraging lightweight diffusion backbones, to further improve efficiency for real-time deployment.

## 5. Conclusions

In this study, we introduced ADZUS, a zero-shot, unsupervised segmentation model leveraging self-attention diffusion mechanisms to address the challenges in biomedical image segmentation. The model’s ability to deliver high-quality segmentation results without the need for annotated datasets marks a significant advancement in the field, especially in scenarios where labeled data is scarce or difficult to obtain. ADZUS demonstrated competitive or superior performance across a variety of medical imaging tasks, including skin lesion segmentation, diabetic foot ulcer or any wound segmentation, chest X-ray infection segmentation, and white blood cell segmentation. These results underscore its versatility and robustness across diverse imaging modalities. The success of ADZUS highlights the potential of self-attention diffusion models to transform medical image analysis by minimizing reliance on extensive labeled datasets. This innovation enhances the accessibility of advanced segmentation tools in resource-constrained settings and paves the way for broader applications in clinical and diagnostic workflows. Future research will focus on expanding ADZUS’s capabilities to additional medical imaging domains, exploring integration with real-time diagnostic systems, and advancing toward a more autonomous segmentation framework. This next phase will enable ADZUS to recognize, label and segment specific zones of interest requested by clinicians, such as infections or abnormal regions, leveraging its learned attention patterns. The promising results of this study suggest that ADZUS could evolve into an intelligent, semi-supervised system capable of automatic region identification and preliminary labeling, thereby enhancing its adaptability for clinical applications while maintaining interpretability and expert oversight.

## Figures and Tables

**Figure 1 bioengineering-12-01036-f001:**
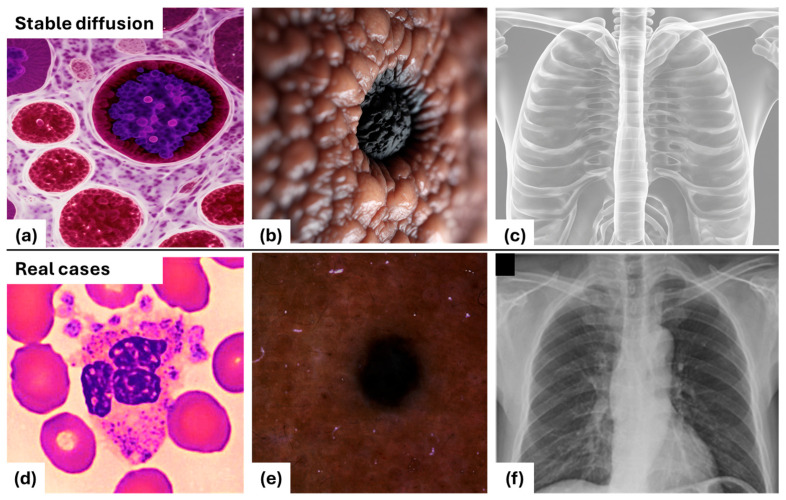
Comparison of Stable Diffusion-generated (**a**–**c**) and real medical images (**d**–**f**) [21,22,23,24,25,26,27,28].

**Figure 2 bioengineering-12-01036-f002:**
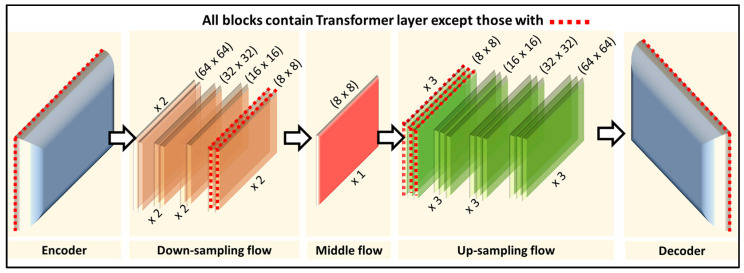
Schematic of the stable diffusion configuration used in ADZUS model, consisting of 16 blocks, each containing transformer layers that produce a 4d self-attention tensor at various resolutions.

**Figure 3 bioengineering-12-01036-f003:**
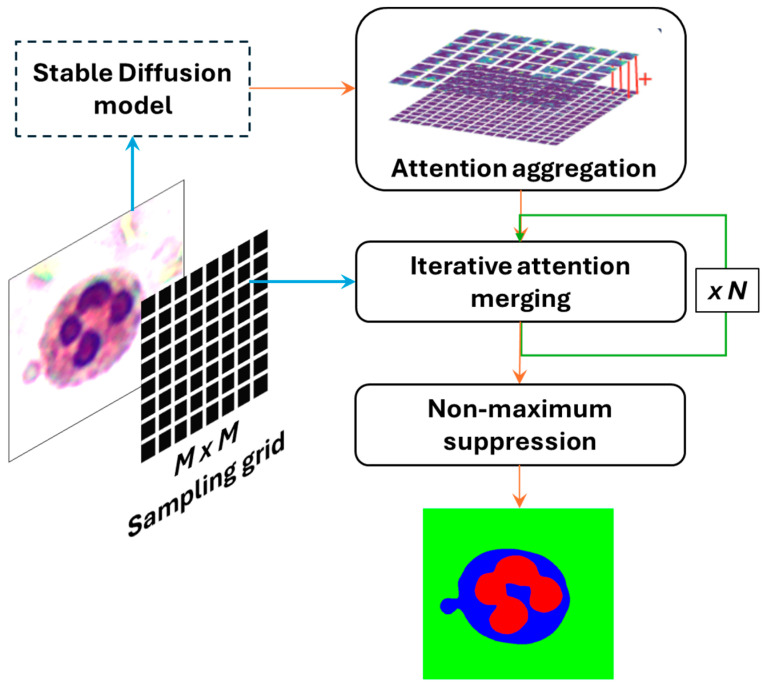
Flowchart of ADZUS model.

**Figure 4 bioengineering-12-01036-f004:**
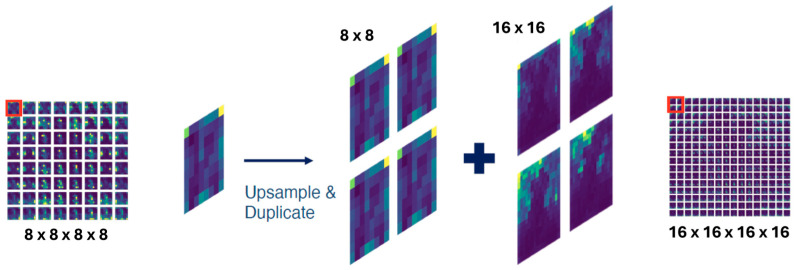
Illustration of Attention Aggregation. A lower-resolution attention map is upsampled and duplicated to match the higher-resolution maps’ receptive field [32].

**Figure 5 bioengineering-12-01036-f005:**
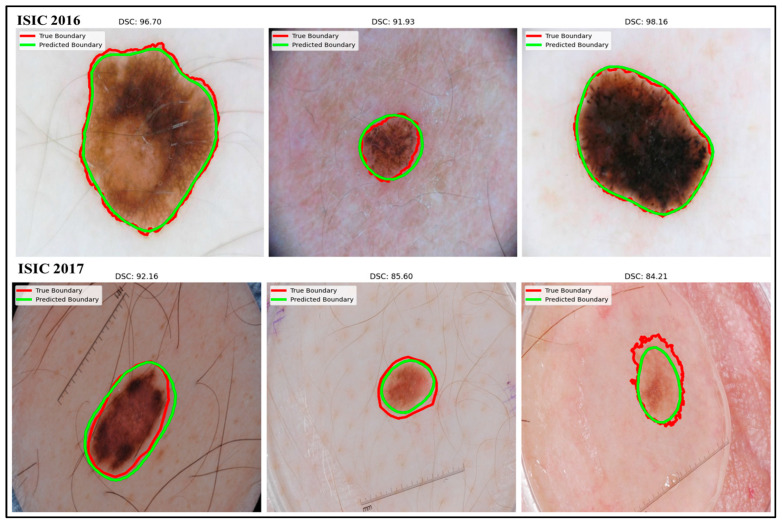
Sample of the qualitative evaluation of ADZUS model on ISIC 2016 and ISIC 2017 datasets: visual comparison of predicted boundaries (green) against true boundaries (red) with corresponding DSC score.

**Figure 6 bioengineering-12-01036-f006:**
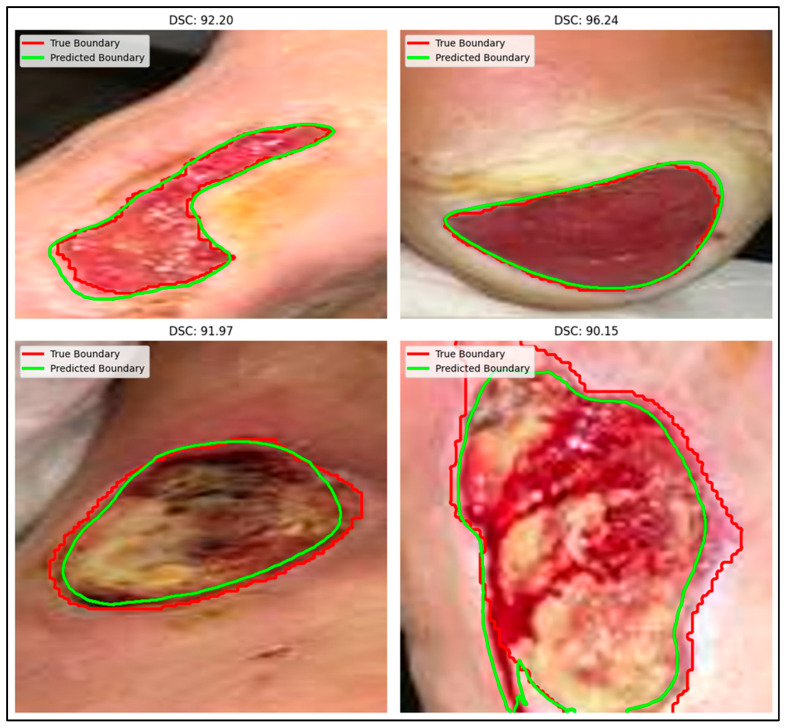
Segmentation results on the chronic wound dataset [37]: original boundaries (red) and predicted boundaries (green) displayed on cropped images for enhanced visualization.

**Figure 7 bioengineering-12-01036-f007:**
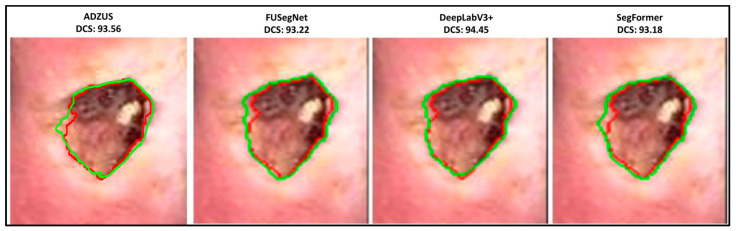
Sample comparison of wound segmentation results: ADZUS vs. benchmark models on chronic wound dataset [23], showing original boundaries (red) and predicted boundaries (green) on cropped images.

**Figure 8 bioengineering-12-01036-f008:**
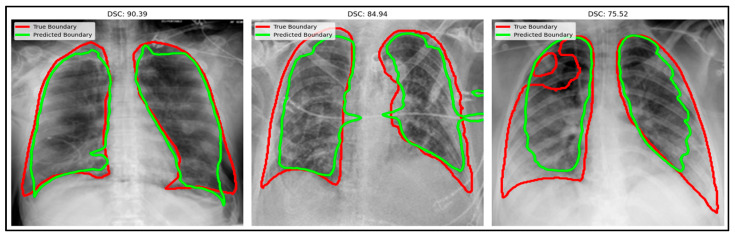
Sample comparison of X-rays segmentation results: original boundaries (red) and predicted boundaries (green).

**Figure 9 bioengineering-12-01036-f009:**
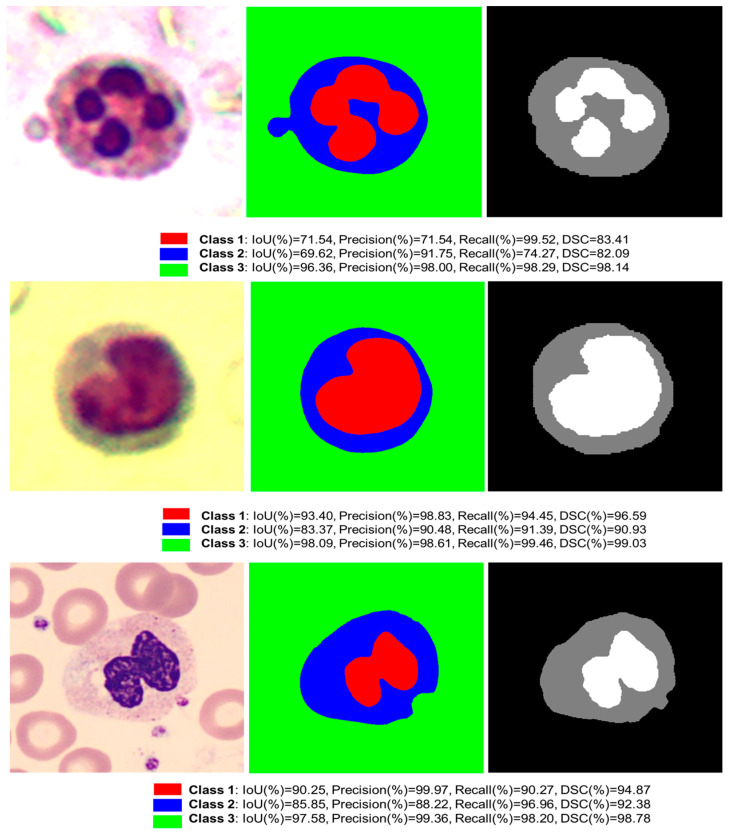
Sample qualitative results of ADZUS on WbcMSBench dataset [26] for multi-class white blood cell segmentation. The figure illustrates the input microscopic images (**left**), ADZUS-predicted segmentation masks (**center**), and ground truth annotations (**right**) for three white blood cell classes.

**Table 1 bioengineering-12-01036-t001:** Performance comparison of ADZUS and state-of-the-art segmentation models on ISIC challenge datasets (2016–2017).

Year	Model	DSC (%)	IoU (%)
ISIC 2016 [21]	U-Net implementation (Top evaluation) [21]	91.0	84.3
	**ADZUS**	**92.9**	**86.8**
ISIC 2017 [22]	Multi-task Deep Learning Model [33]	N/A	72.4
	RECOD Titans Ensemble Approach [34]	N/A	79.3
	Fully Convolutional Network (FCN-AlexNet) [35]	81.9	N/A
	EfficientNet-Based Model [36]	88.0	80.7
	**ADZUS**	**88.7**	**80.0**

**Table 2 bioengineering-12-01036-t002:** Performance comparison of ADZUS and state-of-the-art models in wound segmentation based on DSC and IoU metrics.

Model	DSC (%)	IoU (%)
DDRNet [39]	73.13	57.64
Swin-Unet [39]	88.46	79.30
SegFormer-b5 [42]	91.06	83.58
DeepLabV3Plus [40,41]	92.00	85.19
LinkNet-EffB1 + UNet-EffB2 [38,39]	92.07	85.51
FUSegNet [39]	92.70	86.40
**ADZUS**	**91.98**	**86.68**

**Table 3 bioengineering-12-01036-t003:** Performance comparison of ADZUS and state-of-the-art models in X-rays segmentation based on evaluation metrics.

Model [26]	IoU (%)	Precision (%)	Recall (%)
Resnet18	62.7	74.1	82.4
Resnet50	62.0	73.8	81.0
Efficientnet-b0	63.3	75.3	81.5
Mobilenet_v2	63.1	73.9	82.7
Densenet121	64.7	76	82.6
**ADZUS**	**66.3**	**77.9**	**83.1**

**Table 4 bioengineering-12-01036-t004:** Performance comparison of ADZUS and state-of-the-art models in white blood cell segmentation based on evaluation metrics.

Model [26]	IoU (%)	Precision (%)	Recall (%)
Resnet18	93	96.1	96.6
Resnet50	93.1	96.2	96.6
Efficientnet-b0	93.7	96.5	96.8
Mobilenet_v2	92.6	95.9	96.3
Densenet121	93.8	96.6	96.9
**ADZUS**	**93.3**	**96.5**	**96.7**

## Data Availability

The original data presented in the study are openly available at reference 20–22.

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
