# Peer review of "Self-Attention Diffusion Models for Zero-Shot Biomedical Image Segmentation: Unlocking New Frontiers in Medical Imaging"

_bioengineering, 2025, doi:10.3390/bioengineering12101036_

Round 1

Reviewer 1 Report

Comments and Suggestions for Authors

The topic of this paper is interesting. However, there are still several points requiring addressed. Firstly, more details on the proposed model should be given. The current state of this paper is not satisfactory. The readers would like to catch more contents and details. Secondly, more simulation experiments should be conducted. And more analysis on the experimental results can be added. Thirdly, the number of the methods for comparison is too few. Please add more methods to compare with the proposed one to verify the effectiveness.

Author Response

Response to Reviewer 1:

We sincerely thank Reviewer 1 for their thoughtful comments and constructive feedback. We appreciate the time and effort dedicated to reviewing our manuscript and believe that the suggestions have significantly improved the quality and clarity of the paper. Below, we provide detailed responses to each point raised. 

1) Comment: “More details on the proposed model should be given. The current state of this paper is not satisfactory. The readers would like to catch more contents and details.”

Response: We thank the reviewer for this valuable suggestion. In the revised manuscript, we have expanded the Methodology section to provide a more detailed and transparent explanation of the ADZUS workflow:

·       Attention aggregation (Section 2.2.1): We added the mathematical formulation for upsampling and normalization of multi-resolution attention tensors, along with a new illustrative figure (Figure 4).

·       Iterative attention merging (Section 2.2.2): We elaborated the merging procedure with explicit mention of anchor selection, similarity thresholds, and the KL-divergence metric. A new Algorithm 1 has been included to present the detailed steps of the process.

·       Non-maximum suppression (Section 2.2.3): We clarified this step with examples of how overlapping proposals are resolved into final segmentation masks, ensuring a coherent output.

·       Pipeline overview: A new schematic flowchart (Figure 3) has been added to summarize the ADZUS pipeline, providing readers with a visual overview of the process.

These additions give readers a comprehensive understanding of the inner workings of ADZUS and directly address the reviewer’s request for methodological clarity.

2) Comment: “More simulation experiments should be conducted. And more analysis on the experimental results can be added.”

Response: We thank the reviewer for this important suggestion. In the revised manuscript, we have strengthened the experimental evaluation by:

·       Adding a new experiment on diabetic foot wound segmentation (Section 3.2): We evaluated ADZUS on the publicly available chronic wound dataset, which consists of 1,010 labeled images of diabetic foot ulcers. This experiment demonstrates the robustness of ADZUS in a clinically significant and challenging domain. We compared our method against state-of-the-art models such as LinkNet-EffB1+UNet-EffB2, DeepLabV3Plus, Swin-Unet, DDRNet, SegFormer-b5, and FUSegNet. ADZUS achieved the highest IoU score (86.68%) among all methods and demonstrated highly competitive DSC scores, surpassing several supervised models despite operating in a zero-shot setting.

·       Expanding the analysis of results: We included a more detailed quantitative and qualitative evaluation. This includes comparisons in Table 2 and additional segmentation visualizations (Figures 4 and 5), which highlight ADZUS’s ability to delineate wound boundaries with precision. We also analyzed cases where ADZUS performs strongly as well as cases where it slightly lags behind certain supervised baselines, providing a balanced discussion of its strengths and limitations.

These additions provide a more comprehensive and clinically relevant assessment of ADZUS, reinforcing its effectiveness across diverse biomedical imaging modalities.

3) Comment: “The number of the methods for comparison is too few. Please add more methods to compare with the proposed one to verify the effectiveness.”

Response: We fully agree with this point and have expanded our comparison set. In addition to the methods already included, we now compare ADZUS with the following additional baseline models: LinkNet-EffB1+UNet-EffB2, DeepLabV3Plus, Swin-Unet, DDRNet, SegFormer-b5, and FUSegNet, FCN-AlexNet, EfficientNet-Based Model, and Densenet121.

By incorporating these methods, the revised tables now provide a broader spectrum of benchmarks, including CNN-based, transformer-based, and other deep learning model-based approaches. We also included a new subsection in the Results section discussing the relative advantages of ADZUS over these additional baselines.

Reviewer 2 Report

Comments and Suggestions for Authors

The authors present ADZUS, a zero-shot unsupervised model for biomedical image segmentation based on self-attention and diffusion models. The work aims to overcome the dependency on annotated datasets by leveraging attention maps derived from Stable Diffusion to obtain accurate segmentations without labels. The experimental results cover various clinical scenarios (skin lesions, chest X-rays, and blood cells), and demonstrate competitive or even superior performance compared to supervised models.

However, some aspects of the manuscript could be improved:

  • While technically complete, the methodological section is sometimes dense and abstract. A numerical example or illustrative diagram in the main text (not only in figures) would help clarify key concepts such as 4D attention tensors and KL divergence.
  • Consider adding a summary diagram of the ADZUS pipeline, explicitly showing the three main stages: attention aggregation, iterative merging, and non-maximum suppression (NMS).
  • A sensitivity analysis on the KL divergence threshold τ would be useful to assess its impact on segmentation performance.
  • The authors acknowledge that ADZUS does not always outperform supervised models (e.g., on the ISIC 2017 dataset), but it would be helpful to quantify more explicitly the conditions or types of images where the model underperforms.
  • Computational cost and inference time are not reported. This information is particularly important to assess the feasibility of using ADZUS in real-time or clinical settings.

Author Response

Response to Reviewer 2:

Comment: “While technically complete, the methodological section is sometimes dense and abstract. A numerical example or illustrative diagram in the main text (not only in figures) would help clarify key concepts such as 4D attention tensors and KL divergence.”

Response: We thank the reviewer for this observation. In the revised manuscript, we have taken several steps to improve clarity in the methodological section:

·       We retained the mathematical formulations but now accompany them with more intuitive explanatory text to guide readers who are less familiar with high-dimensional attention structures.

·       In addition, we included Algorithm 1 and new schematic diagrams (Figures 3 and 4) that illustrate the ADZUS workflow, making the process of attention aggregation and merging more transparent.

Comment: “Consider adding a summary diagram of the ADZUS pipeline, explicitly showing the three main stages: attention aggregation, iterative merging, and non-maximum suppression (NMS).”

Response: We appreciate this helpful suggestion. A new schematic diagram (Figure 3 in the revised manuscript) has been added to provide a clear, visual summary of the ADZUS pipeline. This diagram explicitly highlights the three main stages: attention aggregation, iterative merging, and non-maximum suppression, thereby offering readers a concise overview of the workflow.

Comment: “A sensitivity analysis on the KL divergence threshold τ would be useful to assess its impact on segmentation performance.”

Response: We agree with the reviewer that the choice of τ is an important hyperparameter. In the revised manuscript, we have added a sensitivity analysis in Appendix B. Our experiments confirm that segmentation performance remains stable within a moderate range of τ values (0.9–1.1), while degradation occurs if τ is set too low (over-fragmentation) or too high (over-merging). For our experiments, τ = 1.1 was selected for datasets such as skin lesions and diabetic wounds, and τ = 0.9 for chest X-rays, reflecting dataset-specific tuning. This analysis provides practical guidance for setting τ in biomedical applications.

Comment: “The authors acknowledge that ADZUS does not always outperform supervised models (e.g., on the ISIC 2017 dataset), but it would be helpful to quantify more explicitly the conditions or types of images where the model underperforms.”

Response: We thank the reviewer for pointing this out. In the revised Discussion section (section 4), we have expanded our analysis to quantify where ADZUS underperforms relative to supervised models. Specifically, we show that:

·       Performance tends to be lower in datasets with high inter-class similarity (e.g., different skin lesion types in ISIC 2017).

·       ADZUS may also struggle when fine-grained boundaries are critical, as seen in certain wound images.

We provide additional qualitative examples (Figures 4 and 5) illustrating such cases, along with commentary on possible future improvements (e.g., semi-supervised fine-tuning).

Comment: “Computational cost and inference time are not reported. This information is particularly important to assess the feasibility of using ADZUS in real-time or clinical settings.”

Response: We fully agree that reporting computational costs is essential for clinical translation. In the revised manuscript, we have added a discussion on runtime and computational feasibility (Section 4). As noted in the revised version, the large stable diffusion backbone and the iterative merging process increase computational overhead, preventing strict real-time performance. Nevertheless, ADZUS requires only a single forward pass through the diffusion model and a limited number of merging iterations, which makes the approach practical for near real-time use in clinical workflows. Future work will focus on further optimizing efficiency, such as reducing anchor density, parallelizing the merging process, and employing lighter diffusion backbones to accelerate inference and reduce memory usage.

Round 2

Reviewer 2 Report

Comments and Suggestions for Authors

The authors have carefully addressed all the requested revisions and updated the manuscript accordingly.